Skin temperature and reproductive condition in wild female chimpanzees

Dezecache Guillaume guillaume.dezecache@gmail.com 1 2
Wilke Claudia 2 3 4
Richi Nathalie 1 2
Neumann Christof 1
Zuberbühler Klaus 1 2 3
1 Institute of Biology, Université de Neuchâtel , Neuchâtel , Switzerland
2 Budongo Conservation Field Station , Masindi , Uganda
3 School of Psychology and Neuroscience, University of St. Andrews , Fife , United Kingdom
4 Department of Psychology, University of York , York , United Kingdom
Hopper Lydia
Electronic publication date: 2017 Dec 5
Publication date: 2017
Volume: 5
Electronic Location ID: e4116
Received 2017 Jun 21; Accepted 2017 Nov 10
Copyright: ©2017 Dezecache et al.
Copyright year: 2017
Copyright holder: Dezecache et al.
License: This is an open access article distributed under the terms of the Creative Commons Attribution License, which permits unrestricted use, distribution, reproduction and adaptation in any medium and for any purpose provided that it is properly attributed. For attribution, the original author(s), title, publication source (PeerJ) and either DOI or URL of the article must be cited.
License URL: https://creativecommons.org/licenses/by/4.0/

Keywords: Infra-red thermography, Skin temperature, Wild chimpanzees, Pregnancy

Funding: European Union’s Seventh Framework Programme 283871 Budongo Conservation Field Station The research was supported by a Fyssen fellowship awarded to Guillaume Dezecache, and has received funding from the European Union’s Seventh Framework Programme for research, technological development and demonstration under grant agreement (No 283871). We are grateful to The Royal Zoological Society of Scotland for providing core funding to the Budongo Conservation Field Station. The funders had no role in study design, data collection and analysis, decision to publish, or preparation of the manuscript.

==============================
Infrared thermal imaging has emerged as a valuable tool in veterinary medicine, in particular for evaluating reproductive processes. Here, we explored differences in skin temperature of twenty female chimpanzees in Budongo Forest, Uganda, four of which were pregnant during data collection. Based on previous literature in other mammals, we predicted increased skin temperature of maximally swollen reproductive organs of non-pregnant females when approaching peak fertility. For pregnant females, we made the same prediction because it has been argued that female chimpanzees have evolved mechanisms to conceal pregnancy, including swellings of the reproductive organs, conspicuous copulation calling, and solicitation of male mating behaviour, to decrease the infanticidal tendencies of resident males by confusing paternity. For non-pregnant females, we found slight temperature increases towards the end of the swelling cycles but no significant change between the fertile and non-fertile phases. Despite their different reproductive state, pregnant females had very similar skin temperature patterns compared to non-pregnant females, suggesting little potential for males to use skin temperature to recognise pregnancies, especially during maximal swelling, when ovulation is most likely to occur in non-pregnant females. We discuss this pattern in light of the concealment hypothesis, i.e., that female chimpanzees have evolved physiological means to conceal their reproductive state during pregnancy.

Introduction

Infrared thermal imaging (IRT) has emerged as a promising tool for studying animal behaviour. For instance, research employing this methodology has helped cast light on affective processing in a variety of species, including macaques (Macaca mulatta) (Nakayama et al., 2005; Kuraoka & Nakamura, 2011; Ioannou, Chotard & Davila-Ross, 2015), chimpanzees (Pan troglodytes) (Kano et al., 2016; Dezecache et al., 2017) and dogs (Canis lupus familiaris) (Travain et al., 2015; Riemer et al., 2016; Travain et al., 2016). In these studies, IRT has been used to detect changes in emissivity of the skin caused by shifts in blood flow at the body surface, a physiological process controlled by the autonomic nervous system (see (Ioannou, Gallese & Merla, 2014) for a comprehensive review). One important asset of this technique is its non-invasive character, since measurements can be made at a reasonable distance from free-ranging animals and without hindering their on-going natural behaviour.

IRT has also been used as a non-invasive technique to study reproductive processes of animals (Cilulko et al., 2013). For example, Hilsberg-Merz (2008) noticed that female Asian elephants (Elephas maximus) and Black rhinoceroses (Diceros bicornis) showed increased surface temperature in the area of their reproductive organs during oestrus, a pattern associated with increased attractiveness to males. In pigs, vulvar skin temperatures were higher during oestrus compared to dioestrus (Sykes et al., 2012), a pattern related to increased blood flow towards the vulva due to oestrogen secretion in the ovarian follicles (Stelletta et al., 2013). Heightened temperature of the area of the reproductive organs can be used as a tool to detect oestrus in cows, with vulva temperature peaks around 24 h before ovulation, followed by a slight decrease towards ovulation (Talukder et al., 2014). Furthermore, it has been suggested that thermo-reception may constitute a sensory strategy used by males in mammal species in order to detect ovulation in females (Hilsberg-Merz, 2008), possibly in conjunction with other cues.

In sum, most current research using IRT has been performed on domestic and farm mammals, usually in the context of veterinary diagnosis (Cilulko et al., 2013), with only little systematic use in studying natural patterns, including sexual behaviour.

Here, we were interested in the skin temperature of wild female chimpanzees (Pan troglodytes) throughout the swelling cycle. The swelling cycle can be defined as the cyclical change during which a female’s anogenital region gets progressively swollen and increases in size before reaching maximal tumescence, followed by detumescence, whereby the swelling area shows a rapid decrease in size and the skin becomes loose (Wallis, 1992; Deschner et al., 2004). In chimpanzees and other catarrhine species living in multi-male societies, females tend to develop exaggerated anogenital swellings during the periovulatory phase of their menstrual cycles (Nunn, 1999). There has been considerable debate on the biological function of exaggerated swellings, particularly in regard to whether swelling size, or tumescence, constitutes a reliable indicator of fertility. In chimpanzees, males show most interest in females during maximal tumescence (Wallis, 1992; Deschner et al., 2004) when ovulation is most likely to occur (Deschner et al., 2004). Gradual swelling tumescence is caused by increased oestrogen concentrations, while its sudden decrease is caused by increased progesterone secretion (Graham et al., 1972; Emery & Whitten, 2003; Emery Thompson, 2005). However, swelling size is only a probabilistic indicator of fertility in chimpanzees, because maximal swelling can last up to 21 days with a mean of about 13 days and thus maximal tumescence may exceed the phase of peak fertility (Emery Thompson, 2005). This has been interpreted as an evolved female strategy to prolong the period of sexual attractiveness and, by increasing the number of copulations with different males, to confuse paternity (Nunn, 1999; Alberts & Fitzpatrick, 2012). Such a strategy is adaptive in species where males commit infanticide, as is commonly observed across chimpanzee communities (Williams et al., 2008; Goodall, 1986; Nishida & Kawanaka, 1985; Watts & Mitani, 2000; Wilson et al., 2014).

As a first step, we were interested in whether the period of maximum tumescence, if compared to earlier phases, can be identified by a unique temperature profile. In a second step, we were interested in the temperature profiles of pregnant females, who have been suggested to conceal their gestation in an attempt to remain attractive to males (Wallis, 1982). Pregnant chimpanzee females continue to be sexually active and display sexual swellings despite being no longer able to conceive. It has been suggested that the maximal anogenital swelling during pregnancy may account for one fourth of the total time female chimpanzees display maximal tumescence during their reproductive lifetime (Wallis & Goodall, 1993). Pregnant females are, in some instances, even more sexually active than non-pregnant females (Wallis, 1982). One functional explanation of this behavioural pattern is that pregnant females attempt to confuse paternity in the males of their group, which may lower the infanticidal tendencies of males once an infant is born (Wallis & Goodall, 1993).

Based on these findings, we hypothesised that variation in skin temperature may provide cues about the three main reproductive stages of a female, that is to say, pregnancy, oestrus, non-oestrus. However, based on the literature reviewed, female chimpanzees appear to have evolved ways to conceal their true reproductive stage, particularly pregnancy, but also the exact time of ovulation (Townsend, Deschner & Zuberbühler, 2011). If males perceive changes in skin temperature, then selection should favour individuals with skin temperature profiles that do not reveal their reproductive condition.

To address these hypotheses, we investigated skin temperature of female chimpanzees in the Sonso community of Budongo Forest, Uganda. We inspected the thermal patterns of pregnant and non-pregnant females to test the following predictions. First, for non-pregnant females, we expected higher temperatures (particularly of the genital area) during the fertile phase (when females show peak swelling sizes and when sexual proceptivity is highest (Wallis, 1992)) compared to non-fertile phases (when swellings are not maximally inflated and sexual proceptivity is comparably lower (Wallis, 1992)). This was expected because, in farm animals, the skin temperature of the vulva can be influenced by oestrogen secretion through increased blood flow (Stelletta et al., 2013).

Second, if pregnant females follow an evolved strategy to conceal their pregnancy when showing sexual tumescence, we predicted similarity in skin temperature at maximum tumescence between pregnant and non-pregnant females (when male mating efforts are typically concentrated, see Wallis, 1992), compared to earlier swelling stages, when copulation is comparatively rarer and ovulation unlikely.

Methods

Ethical statement

Permission to conduct the study was granted by the Ugandan Wildlife Authority (UWA) (UWA/TDO/33/02) and the Uganda National Council for Science and Technology (UNCST) (NS-475). Ethical approval was given by the University of St Andrews’ ethics committee.

Study site

The study was carried out in the Budongo Forest Reserve, a moist semi-deciduous tropical forest in western Uganda, covering 428 km2 at an altitude of 1,100 m, between 1°35′ and 1°55′N and 31°08′ and 31°42′E (Eggeling, 1947). Data were collected from the Sonso community between November 2011 and May 2012, and between August 2013 and June 2014. Habituation of this community to humans began in 1990, with the majority of individuals (approximate N = 70) well habituated to human observers at the time of the study (Reynolds, 2005).

Materials

Surface skin temperature measurements were taken with a Testo (881–2) thermal imager, which operates between 8 and 14 µm with a thermal sensitivity of <80 mK at 30 °C. Emissivity was set at 0.98, a value typically used for human skin (Steketee, 1973). A telephoto lens was used for all images (9° × 7°/0.5 m). The device emits no light or sound and is thus ideal for working with wild animals.

Pregnancy status

We initially used HCG pregnancy tests (which respond to the presence of >25 mI U/ml human chorionic gonadotropin in the urine, a hormone produced by the placenta about one week after fertilisation). We later decided to estimate pregnancy status depending on the presence or absence of offspring up to 229 days after the recording was taken (assuming a mean gestation period of 229 days in chimpanzees Reynolds, 2005). This was done because it was only possible to perform one or two pregnancy tests for each individual, over a short time frame, so it is possible that some females may have been pregnant temporarily before or after testing. Additionally, a more recent pilot study suggested that HCG pregnancy tests may be unreliable in wild chimpanzees, with pregnant females testing negative around their fifth month of gestation (C Asiimwe, pers. comm., 2016).

As we relied upon the presence/absence of offspring after the average gestation period of chimpanzees, it is possible that some females we designated as ‘not pregnant’ may have been in the early stages of pregnancy when thermal images were recorded, without carrying the pregnancy to term. Stillbirths and miscarriages are difficult to distinguish in the wild (Courtenay & Santow, 1989) (but see Tutin, 1975). Studies in captivity indicate that miscarriages may happen in around 8% of the pregnancies (Courtenay, 1987) and fetal wastage (miscarriage and stillbirths) may occur following 14% of all conceptions (Littleton, 2005).

Thermography data collection

Data collection took place between 07:00 and 16:30 local time. On a given morning, a female was selected as the focal animal and followed throughout the day with IRT photographs taken whenever the individual was in clear view, and photographs taken of surrounding individuals ad libitum. Although we considered all females during data collection, we later excluded 64 images of females with dependent infants (<4 years). Although some of them may have been cycling, we excluded them because of difficulties in determining their reproductive status.

All sampled individuals (20 females, four of which were pregnant during parts of data collection) were sufficiently tolerant to observer presence within 10 m. Each body part (facial region, ears, nose, hands, neck and feet, and swelling/genitals) was sampled only once every half hour with females contributing differently to the dataset (see Table 1). For an image to be taken, the focal had to be within a distance of 15 m (range: less than 1 m–15 m) and less than 5 m above ground. They had to be in unobstructed view, with a body part clearly visible to the observer and in dry conditions (water alters the temperature and emissivity of skin) and not exposed to direct sunlight. Readings were taken from body parts that were exposed and free of hair. For each image, we estimated the distance to the focal individual (in metres). Ambient temperature and humidity data were collected using an electronic recorder, as these may affect infra-red readings. Swelling tumescence was determined by experienced field assistants from stage 0 (absence of tumescence) to stage 4 (maximum tumescence) as judged by the degree of wrinkling (Furuichi, 1987) (adapted for chimpanzees: Townsend, Deschner & Zuberbühler, 2008). Note that the field assistants were blind to the aims and hypotheses of the study.

Table 1 Distribution of images per female (ID) and by reproductive condition (Non-pregnant and Pregnant).

Female ID	Non-pregnant	Pregnant	
AN	9		
HL	3		
IN	2		
JN	61	58	
JT	39		
KA	5		
KL	24	55	
KM	6		
KN	19		
KR	40		
KU	4	72	
KW	52		
KY	38		
ML	38		
MN	6		
NB	100		
NT	100		
OK	68	91	
RH	12		
RS	29		
Total	655	276	

Image analysis

Thermal images were analysed using the Testo IRSoft analysis software. A polygon image selection tool was used on each image to select specific body parts of interest for subsequent thermal analysis. For each selected region of interest, we obtained the mean temperature (see Fig. 1 for example). Two coders (GD and CW) performed the image analysis. To test for inter-observer reliability, we examined measurements of N = 408 thermal images taken by two coders. There was a mean difference of 0.15 ± 0.43 °C (mean difference between the two sets of measurements ± SD) between the two sets of measurements, with high internal consistency (Cronbach’s alpha, α = 0.99), suggesting that our method of calculating the average temperature of an area of interest was consistent.

Figure 1 Example of IRT measurement.

The polygon is drawn around the left ear of the individual which can be seen laying on the ground.

Statistical analysis

We used a linear mixed model with Gaussian error structure and maximum likelihood estimation to assess how swelling stage and pregnancy affected skin temperature of female chimpanzees. In our initial model, we fitted the two-way interaction between swelling stage and reproductive state. This also allowed us to specifically address the possibility that variation in temperature between swelling stages may show different patterns in pregnant and non-pregnant females. In addition, we controlled for ambient temperature, humidity and distance between camera and focal animal. Because some images allowed simultaneous measurement of several body parts, we fitted image ID nested in subject ID as random intercept. In this way, we also accounted for multiple measurements of the same female. We fitted body part as random intercept and in addition, allowed the effects of swelling stage and pregnancy state on skin temperature to vary between body parts by incorporating random slopes for these variables (Barr et al., 2013). Ideally, we would have incorporated similar random slopes for female ID, thereby allowing between subject variation in the effects of swelling and pregnancy on temperature. Unfortunately, we had to forego this step because the resulting model structure was too complex for our data set. Before model fitting, we inspected distributions of variables and transformed them to achieve symmetric distributions (see Supplemental Information). In addition, we scaled all numeric variables to mean = 0 and standard deviation = 1 (Schielzeth, 2010). We checked for homogeneity and normality of model residuals visually and calculated variance inflation factors (Fox & Weisberg, 2010). Neither check indicated serious deviations from modelling assumptions. We then calculated Cook’s distance as a measure for the influence of each single individual in our data set on our model results. Here we found a number of individuals with substantial influence on our results. Specifically, Cook’s distance for all four females in our data set that were pregnant during the study exceeded the critical threshold (c.f. Nieuwenhuis, te Grotenhuis & Pelzer, 2012). Furthermore, we tested full models against our null models (see below) in a leave-one-out fashion to assess the potential influence of single females: we fitted the full and null model with a data set from which one female was excluded per turn. Here, exclusion of one female (OK—see Table 1) led to the full model not being significant (p > 0.05), although the signs of the parameter estimates remained unchanged. Despite this consistency in the direction of our results, the interaction between swelling stage and pregnancy must be interpreted with some caution, as their statistical significance hinges on one individual. However, the signs of our major result concerning the differences between pregnant and non-pregnant females remained consistent, regardless of which female we excluded.

To test the significance of our full model, we built an informed null model, which contained the random effects structure as described above and the three control fixed effects (ambient temperature, humidity, distance). We then tested our full model against this null model using a likelihood ratio test (LRT, Dobson & Barnett, 2008). Similarly, we tested the interaction by comparing the model including the interaction (i.e., the full model) against a model without the interaction (swelling stage and reproductive state as main effects only). We calculated marginal and conditional R2 following Nakagawa & Schielzeth (2013) and Johnson (2014), using the MuMIn package (Bartoń, 2017). We fitted all models in R 3.3.0 (R Core Team, 2016), using the lme4 package (v. 1.1–12, Bates et al., 2015).

From the existing literature (Hilsberg-Merz, 2008; Scolari et al., 2011; Sykes et al., 2012; Talukder et al., 2014), we predicted an increase in temperature throughout the swelling cycle in non-pregnant chimpanzees, with a peak temperature at full tumescence, in particular at the area of the reproductive organs, when females are fertile. Second, and following the hypothesis that female chimpanzees have evolved physiological and behavioural strategies to conceal pregnancy, we predicted pregnant females to show similar patterns of skin temperature changes at peak tumescence when the probability of conception is at its highest for non-pregnant females.

To address the two predictions, we assessed skin temperature associated with swelling and reproductive state, at various body parts, controlling for ambient temperature, humidity and recording distance.

Results

Our full model (containing swelling stage and reproductive state and their interaction plus the control terms: distance, humidity and ambient temperature) was significantly different from the null model (containing the control terms only) (LRT: χ92=19.48, p = 0.0214, Rm2=0.33, Rc2=0.90, Table 2). Concerning our variables of interest—reproductive state and swelling stage—we found that the model containing the interaction between the two was significantly different from a model from which the interaction term was removed (LRT: χ42=9.72, p = 0.0455). Thus, our results indicate that skin temperature showed greater variability in pregnant females (N = 4) than non-pregnant females (N = 20). Generally, pregnant females had lower skin temperatures than non-pregnant females when deflated and during smaller swelling stages (stages 0–2, Fig. 2), of less than 1 °C overall. This pattern changed later in the cycle, with pregnant females having higher skin temperature compared to non-pregnant females (stage 3, Fig. 2), again with a magnitude of less than 1 °C. During maximum tumescence (swelling stage 4), temperatures of non-pregnant and pregnant females appeared most similar compared to all other swelling stages. This was true for all body parts measured (Figs. 3 and 4).

Table 2 Results of full model.

Reference level for pregnancy was ‘Non-pregnant’ and for swelling stage ‘0’. t values for main effects comprised in interactions are omitted.

	Parameter estimate	Standard error	t value	
Intercept	168.15	11.44	14.70	
Swelling stage 1	−7.44	6.91		
Swelling stage 2	−13.94	7.77		
Swelling stage 3	−3.72	5.38		
Swelling stage 4	−12.26	6.52		
Pregnancy	−11.32	6.33		
Ambient temperature	29.65	2.17	13.69	
Distance	−4.66	1.38	−3.37	
Humidity	−4.89	2.21	−2.21	
Swelling stage 1: pregnancy	2.36	8.98	0.26	
Swelling stage 2: pregnancy	−19.81	10.82	−1.83	
Swelling stage 3: pregnancy	17.62	10.32	1.71	
Swelling stage 4: pregnancy	9.98	9.30	1.07	

Figure 2 Model results for differences in body surface temperature.

Shown is the interaction between swelling stage and pregnancy status, with model estimates and associated 95% confidence intervals back-transformed to the original scale (for modeling, surface temperature was squared, see Supplemental Information).

Figure 3 Model predictions for surface temperature of female chimpanzees per body part for non-pregnant and pregnant females.

The temperature axis shows values back-transformed to the original scale (for modeling, surface temperature was squared, see Supplemental Information).

Figure 4 Median surface temperature with 25% and 75% quartiles, of female chimpanzees per body part for non-pregnant and pregnant females.

In contrast to Figs. 2 and 3, we show untransformed raw data here (see Supplemental Information).

Finally, and contrary to our predictions, we did not find a clear increase in temperature throughout the swelling cycle (Fig. 2), at the area of the reproductive organs and elsewhere (Fig. 3). There was a modest increase in skin temperature between stages 1–2 and stages 3–4, in both pregnant and non-pregnant females, in particular around (but not restricted to) the vulvar area (Fig. 3). Yet we did not find a clear difference between the skin temperature at maximal swelling (where female proceptivity is higher Wallis, 1992) and reduced swelling stages. Finally, skin temperature seems comparable or even slightly higher during detumescence than at maximum tumescence.

Discussion

Our aim was to assess whether skin temperature may reveal the reproductive state of female chimpanzees, using IRT, a well-developed technique in veterinary medicine, with yet little application so far in the field of behavioural ecology. We compared wild female chimpanzees throughout their swelling cycle and predicted that (i) non-pregnant females should show higher skin temperature when oestral than when anoestral, as well as an increase in skin temperature (with maximal temperature at the end of the swelling cycle, particularly at the area of the reproductive organs), following previous observations in other species (Hilsberg-Merz, 2008; Scolari et al., 2011; Sykes et al., 2012; Talukder et al., 2014). We also predicted that (ii) pregnant females should approximately overlap with the temperature patterns exhibited by non-pregnant females, despite their radically different hormonal state at maximum swelling (stage 4) when males are sexually interested in them. This second prediction is based on the hypothesis that pregnant females have evolved gestation concealing strategies, as they incur an adaptive advantage in concealing pregnancy, to enhance the benefits of paternity confusion by promiscuous mating as long as possible.

Our results were consistent with the first prediction, by showing slightly higher temperatures of the anogenital regions when transitioning from stages 1–2 to 3–4 (Fig. 2), consistent with what has been found in other mammal species (Hilsberg-Merz, 2008; Scolari et al., 2011; Sykes et al., 2012; Stelletta et al., 2013; Talukder et al., 2014). However, this pattern was true for most body parts considered, not solely for the anogenital region (Fig. 3). Although it is hard to directly compare our results to previous studies using IRT because of different definitions and methods, the differences we found in our results are ofsimilar magnitudes (about 1°C) compared to other studies (Scolari et al., 2011; Sykes et al., 2012). In both human and non-human female primates, skin colour and tone can change during the fertile period (Roberts et al., 2004; Dubuc et al., 2009), and such variation may be perceived by males (Higham et al., 2011). Variation in skin colour, presumably caused by blood flow and associated temperature changes, may affect the entire body, including the face, and not be restricted to the periovulatory area.

Regarding our second prediction, we found smaller changes in skin temperature in non-pregnant compared to pregnant females. Crucially, for stage 4, we observed more similar skin temperatures between pregnant and non-pregnant females, compared to the other swelling stages (Fig. 2), a pattern true for all body parts considered (Fig. 3). Our data is therefore consistent with the prediction that, during gestation, chimpanzee females approximate behavioural and physiological cues that characterise non-pregnant females, which functions to conceal their reproductive state. This could be part of an evolved strategy to deceive males by remaining sexually attractive to them and ultimately to confuse paternity and lower infanticide risk after parturition. Alternatively, the overlap in anogenital temperature could be the by-product of physiological mechanisms involved in anogenital swelling. In the context of the concealment hypothesis, it is worth mentioning that pregnant females show less clear transitions between swelling stages compared to non-pregnant females. Instead, swelling sizes appear to change more erratically, and hence a probably simpler means for males would be to attend to gradation of swelling changes. Previous work has already shown that females show irregular swelling patterns during the early stages of pregnancy (Wallis & Goodall, 1993).

Sexual swellings during pregnancy have also been reported in other non-human primates, such as sooty mangabeys (Cercocebus atys) (Gordon et al., 1991) and Barbary macaques (Macaca sylvanus) (Möhle et al., 2005), but in these species the swelling patterns between pregnant and non-pregnant females seem to differ. Whether males are responsive to non-monotonic changes in swelling and, if so, whether this affects their mating behaviour and future infanticidal tendencies, would be worth exploring.

The important question of whether temperature changes are perceivable by males remains unresolved. In humans, thermal discrimination has been investigated in the domain of psychophysics, using tasks where participants are presented with pairs of materials and instructed to choose the cooler of two objects. Thermal discrimination varies with the nature and size of contact sites as well as the baseline temperature of the skin around the contact site (Ho & Jones, 2006). Also, the rate and magnitude of temperature changes play a considerable role in the discrimination of thermal increments (Jones & Berris, 2002; Ho & Jones, 2006). The thenar eminence at the base of the thumb is one of the most sensitive body parts with reported thermal discrimination of less than 0.1 °C (Ho & Jones, 2006). If chimpanzees possess temperature discrimination capacities similar to humans, then males should be able to perceive the temperature changes reported in this study.

It is also worth mentioning that primate males almost certainly use additional signs to make fertility assessments of females during their sexual swelling cycles, particularly olfactory signals (Michael & Zumpe, 1982). In humans (Homo sapiens), female body odour during highly fertile days is preferred by males (Gildersleeve et al., 2012). In other primate species, olfactory cues may also play a role (Ziegler et al., 1993; Converse et al., 1995; Clarke, Barrett & Henzi, 2009), but, to our knowledge, relevant systematic research in chimpanzees has not been conducted (Fox, 1982). Regarding visual signals, skin colour and tone can change during the fertile period in human and non-human primate females (Roberts et al., 2004; Dubuc et al., 2009), and such variation may be perceived by males (Higham et al., 2011). In humans, facial redness has been linked to the vasodilatation caused by oestradiol (Jones et al., 2015), a pattern in blood flow around the facial area that may be associated with increase in skin temperature. Finally, female sexual behaviour itself can reveal the most fertile period of the swelling cycle (Engelhardt et al., 2005). Future studies should also investigate whether non-pregnant vs. pregnant female chimpanzees show reliable differences in those areas.

We acknowledge several limitations to our research. First, we did not find a clear increase in skin temperature when comparing anogenital areas of females in fertile and non-fertile stages. Second, only four pregnant females contributed to our dataset, so it would be important to replicate these findings with a larger sample of pregnant females. Third, we did not collect any hormonal data and were therefore unable to determine the point of likely ovulation. A validation study is necessary and would moreover be feasible in captive settings. Finally, we had no behavioural data to show that males are directly responsive to changes in skin temperature, and can be deceived by pregnant females who may have similar skin temperature profiles as fertile individuals. Such behavioural data could have also helped to control for other potential factors that may have affected skin temperature, such as the amount of physical activity when resisting solicitations from males (Stumpf & Boesch, 2005). Nevertheless, we consider it plausible that males can be affected by pregnant females’ skin temperature, mainly because shifts in blood flow, and their corresponding changes in skin temperature, may affect skin colouration in the face.

In sum, our data appear consistent with the prediction that, during gestation, chimpanzee females approximate skin temperature as well as behavioural and visual cues that characterise non-pregnant females. Yet it is still possible that skin temperature does not constitute a deceptive signal in chimpanzees. Rather, skin temperature may simply be a by-product of the physiological mechanisms driving anogenital tumescence. Furthermore, we offer inconclusive evidence of a thermal pattern associated with fertility. Skin temperature seems to increase throughout the swelling cycle, but with no clear differences in skin temperature compared to when females are anoestral. Our research offers a tentative exploration of changes in skin temperature associated with fertility and pregnancy in wild female chimpanzees, which future research can build on, using IRT to tackle important questions in the field of behavioural ecology.

Supplemental Information

Data S1 Dataset

Raw dataset

Click here for additional data file.

Supplemental Information 1 Script for analysis

Click here for additional data file.

Supplemental Information 2 Read me

Supporting information for the interpretation and use of the raw data and analysis script.

Click here for additional data file.

We are grateful for comments by Lydia Hopper, Katherine Cronin and two anonymous reviewers. We thank the Uganda Wildlife Association and the Uganda National Council for Science and Technology for permission to conduct the study. Our gratitude goes to Geoffrey Muhanguzi, Caroline Asiimwe, Geresomu Muhumuza, Bosco Chandia and Sam Adue for their support in the field. We further thank Cat Hobaiter, Roman Wittig, Dave Perrett, William Paterson, Dominic McCafferty, Ross Whitehead, Amanda Hahn, Brittany Fallon and Zanna Clay for helpful discussions and/or comments on the manuscript.

Additional Information and Declarations

Competing Interests

Author Contributions

Animal Ethics

Data Availability

The authors declare there are no competing interests.

Guillaume Dezecache performed the experiments, analyzed the data, wrote the paper, prepared figures and/or tables, reviewed drafts of the paper.

Claudia Wilke conceived and designed the experiments, performed the experiments, analyzed the data, reviewed drafts of the paper.

Nathalie Richi performed the experiments, analyzed the data.

Christof Neumann analyzed the data, contributed reagents/materials/analysis tools, wrote the paper, prepared figures and/or tables, reviewed drafts of the paper.

Klaus Zuberbühler conceived and designed the experiments, wrote the paper, reviewed drafts of the paper.

The following information was supplied relating to ethical approvals (i.e., approving body and any reference numbers):

Permission to conduct the study was granted by the Ugandan Wildlife Authority (UWA) (UWA/TDO/33/02) and the Uganda National Council for Science and Technology (UNCST) (NS-475). Ethics approval was given by the University of St Andrews’ ethics committee.

The following information was supplied regarding data availability:

The raw data has been uploaded as a Supplementary File.

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
