# Peer review of "Skin temperature and reproductive condition in wild female chimpanzees"

_PeerJ, doi:10.7717/peerj.4116_

## Round 0.1 · original submission · Major Revisions

I have now received reviewers from three experts in your field. As you can see their evaluations of your article are mixed. However, all three raise some serious concerns regarding the theory and execution of your study extending upon the considerable limitations you note yourselves in your Discussion (lines 274-285). With the lack of associated behavioral data, the small sample size, and the lack of reliability testing of the IRT methods, this study feels more like a pilot assessment of the methodology, than an evaluation of the concealed pregnancy hypothesis.

However, given the positive comments provided by Reviewer 1 and Reviewer 3 I would like to give you the opportunity to revise your manuscript for resubmission to PeerJ. I believe that if you are able to provide reliability metrics for your methods, so that the reader could have greater confidence in your results and, by default, the methods, this study would provide greater benefit to the field and for those who are considering using such methods as a non-invasive measurement technique.

In addition to the comments provided by the three reviewers, I have a few points that I would like you to respond to as well:

In your Introduction, the model species that you provide citations for are all captive non-primate species (lines 40-55), yet as Reviewer 3 notes, IRT studies have been conducted both with primate species and with animals in the wild. Please provide discussion of these previous studies, and also please provide greater consideration as to why you think the results found for pigs, rhinos, elephants and cattle should translate to (wild) chimpanzees. Are there data from humans that suggest changes in body temperature (at the genital region or elsewhere) during cycling or pregnancy? See also the comments made by Reviewer 1 to this point as well.

Both Reviewers 1 and 3 question the IRT methods used and request greater clarity on how they compare to previously-used methods. I agree that such a description is essential. Furthermore, and as Reviewer 3 notes, it appears that your IRT methods lack validity and reliability. I understand that it would be difficult to obtain temperature measures from wild chimpanzees using other means, but perhaps the technology could be validated with humans tested in the same location as the chimpanzees (i.e. with the same ambient temperature, humidity etc.) by testing humans both with IRT and a thermometer. Alternatively, to ensure reliability, could you have taken multiple measures of the same individual within a short time period to evaluate the consistency of readings? Without such understanding of how reliable your data are it is difficult to evaluate the results that you report. If possible, I would encourage you to perform reliability testing of the IRT methods – at the very least with humans in a laboratory/university setting if field testing is not possible.

I also agree with Reviewer 3’s concerns about the limitations of your method for recording pregnancy and the certainty of your study subjects’ pregnancy status during the study. I appreciate that it can be difficult to obtain such measures in the field, but you must acknowledge this limitation and also the possibility that not all initially-pregnant females will complete gestation, thus creating false negatives. Also, is it likely to expect changes across the course of pregnancy? Again, without knowing these details the results have reduced impact.

You were only able to include four pregnant females in your study. While you mention this in your Discussion, this point feels obfuscated in its presentation in Table 1. Perhaps in the Methods or Results you could mention explicitly the number of females included in these analyses.

It is also disappointing that your study lacks behavioral data, especially as you note in your Introduction that you would expect changes in the male behavior towards females across their cycles. This also limits any interpretations about pregnancy concealment. Do you have any notes about the number of copulations during this period?

A small note: Please move the description of your aims (lines 185-193) from your Results section to your Introduction or to your Analysis section.

·

Basic reporting

The authors investigated body temperature changes of pregnant and cycling chimpanzee females in the field. They have used a technology new to behavioral ecology to test two clearly stated hypotheses. The first is that surface body temperatures will increase throughout the swelling cycle and peaking at maximum tumescence. The second hypothesis is that the temperature patterns of pregnant females will mimic those of non-pregnant females. The manuscript is well-written, well-structured, and clearly organized raw data and Rmarkdown analyses details have been shared as supplemental information.

I have the following minor comments regarding basic reporting:

1. If females have evolved a strategy to conceal temperature changes as part of a broader strategy to conceal pregnancy, one must assume that males can detect the slight temperature changes that would correspond with pregnancy. Is it logical to assume male chimpanzees would be able to detect such small differences in female skin temperature? Is there any literature that would suggest temperature to be a meaningful, detectable signal to male chimpanzees, or to other species? (The authors do mention that male elephants were more attracted to female elephants with increased skin temperature Hilsberg-Merz 2008 , 
but presumably the attraction could have been due to olfactory or other cues also associated with reproductive state. They also mention male primates responsive to skin color and tone, but not temperature in the Higham et al 2011 reference). I think it is fine to explore this hypothesis as an extension of the concealed pregnancy hypothesis, but wanted to encourage the authors to include more justification for their hypothesis that is based on temperature perception if it is available. If it is not available, this lack of knowledge should be explicitly included in the introduction.

2. Number of cycling and pregnant females should be included in the abstract.

3. Figure 2: Is there a reason why the y-axis does not have equally spaced intervals? These should be equally or clearly labeled on the y-axis as reflecting a specific scale.

Experimental design

The research question is well defined, relevant and meaningful. I have the following minor comments regarding experimental design:

1. What transformations were made to the variables (lines 158-159)? This information should be included in the main text, or reference to the supplemental information should be made here.

2. Has IRT been used before on wild chimpanzees? If so, are the methods stated in paragraph beginning line 109 based on previous studies?

3. How much do the image analysis results depend on the polygon drawing? It seems there should be inter-rater reliability on average temps obtained due to the user-defined polygons.

Validity of the findings

While the data are somewhat inconclusive, the authors have shared important new methodological approaches and been upfront about the limitations of their results. I have the following minor comment about validity of the findings:

1. The authors are very upfront about limitations in their design (paragraph beginning line 274). Should decisions about photographing distance/time/lighting also be included here as variables that may influence findings, given the rarity with which IRT is used in the field?

Additional comments

Additional minor comments:

Line 51: “cattle animals” – reword as cattle, livestock, or farm animals.
Line 42: parenthesis missing around citation
Line 74: “In a first step” – reword as “As a first step”
Line 75: suggest deleting “seemingly strategic”
Line 77: change “which” to “whom”
Line 123: “sampled” refer to sampled with IRT?
Line 125-6: could females with offspring <4 years of age been cycling, even at a low probability? If so, rephrase.
Line 171: “this” effect and “the” interaction – it’s not clear which variables are being referred to
Line 202: is “deflated” a common term for minimal swelling? If not, consider replacing.
Figure 1: It is difficult to discern the content of the images, is there perhaps a standard image available not showing the infrared that could be shown side by side? Alternatively, could some labels be placed on the photos for reference?

Reviewer 2 ·

Basic reporting

This manuscript had several typos and poor English construction in some parts. I would encourage the authors to make sure they carefully review a manuscript before submitting it for publication.

I was never convinced about the purpose of this study. There may well be temperature changes during the reproductive cycle – such as seen in humans and other animals – but it’s not clear why the authors have jumped to the conclusion that any temperature change would serve as a “signal” to males. Has anyone ever proposed that humans use genital temperature as a cue? Or do cattle researchers think bulls use vaginal temperature of cows as signals? I doubt it. I don’t get the significance of temperature in communication.

Also, remember that most of what’s physically causing the swelling of female chimpanzee ano-genitalia skin is the interstitial fluid (mostly water). Some parts of the manuscript seem to imply the swelling is mostly blood. That’s not true.

Experimental design

The authors admit there were several problems with the actual design. I wonder if it would have been best to first set up an experimental design that would test the validity of the technology used in that setting, at those distances, and in those environmental conditions.

Validity of the findings

Mostly this represents negative results - which is OK in this case. Graphic details were well presented.

I appreciate the fact that the authors went to great lengths to describe the shortcomings of the study, but it bordered on overkill - as though it underscored the lack of importance and planning for the study.

Additional comments

Careful observation of female chimpanzee swelling patterns allow for prediction of pregnancy very early on. When observation is made daily (which is possible at Budongo), one can easily map the irregular patterns of early pregnancy and predict delivery date. See the Wallis and Goodall 1993 paper for details on this and further discussion about the value of swelling during pregnancy.

I would recommend carefully assessing the goal of the study. Maybe I'm missing something, but it seems you are suggesting that skin temperature may be being used as a cue for males and there's no logical explanation given for this.

Reviewer 3 ·

Basic reporting

This manuscript was clearly divided into different sections: introduction, methods, results and discussion. Overall, I could follow the prose in each section.

For me, the main drawback was the set up for the rationale of the study. I did follow the logic that based on prior research (40-50), we may expect female chimpanzees’ skin temperature to vary over the reproductive cycle. However, I did not understand the rationale for predicting that pregnant females at maximal tumescence will mimic the temperature profile of non-pregnant females at maximal tumescence in order to confuse paternity. I wonder, is there evidence of this in other species? Reading the authors’ description of a study on elephants and rhinos (41-44), it was not clear whether males were more attracted to females during oestrous specifically due to skin temperature changes, or other signals (I could not access the cited article). If there is evidence that certain species use skin temperature of potential mates to assess reproductive state, it should be discussed in more detail in the introduction. If there is not literature on this topic, why should we assume that male chimpanzees can and do attend to the skin temperature of females and that females evolved a temperature profile to deceive males and confuse paternity?

Some language in the manuscript was ambiguous, causing me confusion:

201-204: “Generally, pregnant females had lower surface temperatures than cycling females when deflated and during smaller swelling stages (stages 0 – 2, Figure 2), of less than 1°C overall. This pattern changed later in the cycle, with pregnant females having higher skin temperature compared to cycling females”

It is confusing to apply the words “cycle” and “cycling” to both pregnant and non-pregnant females. The authors differentiate two groups of females “cycling vs. pregnant females” (11-12, 201-202) but then in the same sentence, mention the cycles of both pregnant and cycling females (203-204). The authors should select unambiguous phrasing for the different female reproductive groups and use it consistently throughout the manuscript (e.g. perhaps non-pregnant females).

Another example of confusing language is referring to pregnant females as non-reproductive:

94: “if pregnant females follow an evolved strategy to conceal their non-reproductive state”

Grammatically speaking, throughout the manuscript there were areas of awkward language (e.g. 40: “strands of research”; 74: “In a first step”; 293: “which has not or seldom been used”). Cleaning up these phrases will allow the reader to concentrate on the research, rather than the grammar.

53: The authors state, “we are not aware of systematic use of this technology on wild animals” but see Thompson et al. 2017, Journal of Thermal Biology, for a study on wild howler monkeys:
http://www.sciencedirect.com/science/article/pii/S0306456516302182

Experimental design

I was pleased to see research aimed at better understanding wild female chimpanzee physiology, as there is much discover in this area. In addition, I appreciate the authors highlighting how infrared thermal imaging is a tool that can be used to study animals noninvasively.

I found the authors’ two study aims to be clearly conveyed:
1. Do non-pregnant females have higher skin temperatures during estrous compared to anoestral phases?
2. Do pregnant females at maximum tumescence have similar skin temperatures compared to non-pregnant females at maximum tumescence?

The authors also included a lot of detail about the statistical methods, which was useful.

On a critical note, I have a concern about study methodology. I question how the authors designated pregnant versus non-pregnant females using the presence or absence of offspring after the average gestation period of chimpanzees elapsed following thermal image collection (114-120). During my own experience conducting pregnancy tests on wild female chimpanzees, pregnancy loss ostensibly occurred during the early months of gestation (based on several consecutive positive pregnancy tests followed by negative tests). Evidence of early pregnancy loss is also mentioned in Emery Thompson (2005, pg 147-148). This means that images of females in early pregnancy could have been included in the non-pregnant female category. I know it is not possible to get around this issue, but at least mentioning it will allow for better assessment of the results by the reader.

Validity of the findings

The authors state: “…we did not find any difference in skin temperature between measures taken from anoestral and oestral stages in cycling females. This is inconsistent with the existing literature, and casts some doubt on the possibility that fertility is associated with a general thermal signature in female chimpanzees” (242-245).

I wonder about validity of infrared thermal imaging, especially in a wild environment. Although the authors control for ambient temperature, humidity, and recording distance in their statistical models (which is good), they do not mention potential drawbacks of infrared thermal imaging.

Some captive studies indicate low reliability of infrared thermal imaging to assess temperatures in primates:

http://www.jzar.org/jzar/article/view/132

http://onlinelibrary.wiley.com/doi/10.1111/j.1600-0684.2007.00214.x/full

It may be useful for the authors to mention whether any steps were taken to assess efficacy of their thermal imaging data. For example, Pizzi et al. (2015) mention examining reliability coefficients.

http://www.jzar.org/jzar/article/view/132


Regarding the discussion, I am not convinced that because skin temperatures for pregnant and non-pregnant females were most similar during maximal tumescence, that skin temperature is necessarily part of a female strategy to remain sexually attractive and confuse paternity. The authors do not explore any alternative explanations. For example, it is possible that skin temperature is not a deceptive signal in chimpanzees, but rather, something about being maximally tumescent increases body temperature regardless of pregnancy status. There is much research on wild chimpanzee females documenting changes that occur during maximal tumescence including increased: copulation rates (Watts 2007), male sexual competition over females (Emery Thompson and Wrangham 2008), and female resistance to male solicitation (Stumpf and Boesch 2005). Could increased activity and social upheaval lead to small temperature increases during maximal tumescence? It is certainly possible. Because the authors’ dataset is modest, and they do not present the reader with clear evidence that skin temperature is a signal in animals, the discussion is not very compelling. In particular, the use of the Hiramatsu et al. 2017 citation (285) as support seems like a reach, as it is about facial coloration in rhesus macaques, as assessed by humans. The manuscript could be improved by focusing the discussion more closely on the data presented and acknowledging competing explanations for the observed temperature patterns observed.

---

## Round 0.2 · Minor Revisions

Thank you for making substantial improvements to your article and for responding to both my and the reviewers' feedback. I am satisfied that you have addressed the majority of the concerns raised previously. However, before I can accept your article for publication, there are a few more minor edits that I request you address. Two of the three reviewers who originally reviewed your article still have some outstanding concerns. Furthermore, both these reviewers note that your article has a number of typos and grammatical errors. To address these, I encourage you to thoroughly proof read your article as PeerJ does not provide such copy editing.

·

Basic reporting

No comment.

Experimental design

No comment.

Validity of the findings

No comment.

Additional comments

In this revision, the authors have addressed the minor concerns that I had with the original submission.

Reviewer 2 ·

Basic reporting

This version is much improved. The basic reporting and article structure are fine - and it appears most of the relevant literature have been included.

See below for more details.

Experimental design

I find no problems with the experimental design, although it may be useful to carefully review the methods and results section with the editorial staff; some of the wording is awkward and less clear than could be achieved with some additional effort.

Validity of the findings

I'm pleased with the new version. Many of my concerns about the initial version have been addressed. In particular, the authors have toned down their suggestions about the "signalling" potential of temperature of the genitalia.

One thing that concerned me last time that I failed to bring up: If there is anything about temperature that helps communicate information to the males and influence their behavior, we should be just as interested in how the temperature change influences the FEMALES' behavior. To imply that something going on in a female is only important to study as it relates to males is short-sighted. In other words, this is akin to suggesting that the anogenital swelling of a female is only about communicating to the male. We wouldn't ignore the very obvious tactile stimulation being experienced by the female herself - and we have plenty of data showing very proceptive behavior by females during times of swelling.

Another thing to consider is that, though anogenital tumescence occurring during pregnancy is very likely a phenomenon that helps with sociosexual relationships, we can't say it is ONLY about confusing paternity. There are studies from captive conditions that confirm sexual attractivity and receptivity during pregnant (while swollen) - but those same studies show a difference in males' grooming and genital inspection rates. So, actual masking of pregnancy may or may not be going on. In other words, they are being treated slightly differently from the non-pregnant females - and still having affiliative behaviors (including sex) shown to them.

Additional comments

Though the manuscript is much improved and I support publication, there are still a fair number of typos and basic sentence structure problems throughout. Much of this will be caught by careful proofreading by the editorial staff.

Some specific areas of concern (these are just some examples):

line 37: extra parenthesis should be deleted
line 57: re-word this bit. You refer to a "swelling cycle" before explaining what that is. for those unfamiliar with chimpanzees, they need an introduction.
line 63: be careful when referring to swelling "size." It's not really about the physical size but the level of tumescence (and lack of "wrinkles").
line 87 (and elsewhere): You may want to avoid using "oestrus" - and just stick with fertile. Many researchers argue that the term oestrus should be limited to those animals that have an extremely discrete period of sexual receptivity that is directly related to the ovulatory period. Clearly, chimpanzees exhibit interest in sexual behavior outside the time of ovulation (though they DO show higher interest while the swelling is present). Thus, I would caution how you use the term and - if you're going to use it, at least include a short discussion/disclaimer about the term.
line 97, 99: extra parenthesis
line 111: missing end period
line 279: You mean to write "during pregnancy"
line 326: Period is missing after "fertility" --- and similar missing punctuation occurs throughout the manuscript.

In other words, provide a careful proofreading by multiple people to catch these minor mistakes and address the other issues I've mentioned here. At that point, I think it will be suitable for publication.

Reviewer 3 ·

Basic reporting

Overall, the manuscript is improved compared to the previous submission. Upon the advice of the editor/reviewers, the authors have provided clarity in several areas, including the sample size of females (both pregnant and not), as well as some of the background literature cited. I also found that this version of the manuscript was easier to read throughout all sections, as the authors made the recommended changes regarding sentence clarity and grammar. There were some minor grammar/typographical errors, which I note under "General Comments" as they did not greatly affect the manuscript's readability.

Experimental design

I appreciate the authors response to the critique about the limitations of designating pregnant versus non-pregnant females using presence/absence of offspring after the average gestation period of chimpanzees elapsed following thermal image collection. However, I do not think the possibility of pregnant females being included in the non-pregnant female dataset is explicitly stated (lines 127-136). The authors have mainly added a justification for not relying on pregnancy tests, which is not the core issue. A clear sentence would be something like, “As we relied upon the presence/absence of offspring after the average gestation period of chimpanzees elapsed following image collection, it is possible that some females we designated as 'not pregnant', may have been in the early stages of pregnancy when thermal images were recorded, and then subsequently did not carry the pregnancy to term.”

Validity of the findings

While I do think the authors were more cautious about the discussion of their findings in light of the pregnancy concealment hypothesis in this version of the manuscript, my main critique stands. The authors do not explore any alternative hypotheses for why pregnant and non-pregnant maximally tumescent female chimpanzees have similar temperature profiles.

I copy and paste what I stated in my initial review, as it still is applicable:

“The authors do not explore any alternative explanations (for their findings). For example, it is possible that skin temperature is not a deceptive signal in chimpanzees, but rather, something about being maximally tumescent increases body temperature regardless of pregnancy status.”

Considering that a) this study has a very small sample size of pregnant females b) the temperature changes recorded throughout the study are very small in magnitude, and c) there is no evidence that male chimpanzees can detect changes in skin temperature of females, it seems like a huge misstep to completely ignore alternative explanations for the authors’ findings.

It is certainly more simplistic (Occam’s razor) to conclude there is something about being maximally tumescent that increases body temperature regardless of pregnancy status, rather than arguing that female chimpanzees have evolved a strategy to mimic tiny temperature changes that males can somehow attenuate to.

Additional comments

The authors have done a good job responding to and addressing the previous critiques. Moving forward, the most important thing the authors can do is acknowledge alternative explanations for their results (other than the pregnancy concealment hypothesis). This can be done easily in the discussion.

Secondly, minor changes include being explicit about the designation of pregnancy status, and finally, cleaning up the grammatical and typographical errors.

34: or dogs -> and dogs

35: deployed -> employed or utilized

56: may be useful to add “wild” before “female chimpanzees”

150: fur -> hair

226: “surface temperatures” -> skin surface?

235: body temperature -> skin surface temperature

288-294: Very confusing section. What are these studies about? How were they conducted? Needs more explanation.

314: The line about difficulty getting hormonal samples is a bit confusing to me. There’s plenty of hormonal work from Emery Thompson, Deschner, Crockford and others (at Budongo and elsewhere). Perhaps it was not feasible for this study’s timeframe or budget?

326: "skin" is capitalized and should not be

---

## Round 0.3 · accepted · Accept

I believe that you have thoroughly addressed all of the reviewers' outstanding concerns. Obtaining feedback from a native speaker has also improved the clarity of the phrasing throughout. It is my pleasure to accept your article for publication in PeerJ.